

# Co-benefits of global and regional greenhouse gas mitigation on U.S. air quality in 2050

Yuqiang Zhang[1], Jared H. Bowden[1], Zachariah Adelman[1,2], Vaishali Naik[3], Larry W. Horowitz[4], Steven J. Smith[5], J. Jason West[1]

[1]Environmental Sciences and Engineering Department, University of North Carolina at Chapel Hill, Chapel Hill, NC 27599
[2]Institute for the Environment, University of North Carolina at Chapel Hill, Chapel Hill, NC 27599
[3]UCAR/NOAA Geophysical Fluid Dynamics Laboratory, Princeton, NJ 08540
[4]NOAA Geophysical Fluid Dynamics Laboratory, Princeton, NJ 08540
[5]Joint Global Change Research Institute, Pacific Northwest National Laboratory, College Park, MD 20740

*Correspondence to*: J. Jason West (jjwest@email.unc.edu)

**Abstract.** Policies to mitigate greenhouse gas (GHG) emissions will not only slow climate change, but can also have ancillary benefits of improved air quality. Here we examine the co-benefits of both global and regional GHG mitigation on U.S. air quality in 2050 at fine resolution, using dynamical downscaling methods, building on a previous global co-benefits study (West et al., 2013). The co-benefits for U.S. air quality are quantified via two mechanisms: through reductions in co-emitted air pollutants from the same sources, and by slowing climate change and its influence on air quality, following West et al. (2013). Additionally, we separate the total co-benefits into contributions from domestic GHG mitigation versus mitigation in foreign countries. We use the WRF model to dynamically downscale future global climate to the regional scale, the SMOKE program to directly process global anthropogenic emissions into the regional domain, and we provide dynamical boundary conditions from global simulations to the regional CMAQ model. The total co-benefits of global GHG mitigation from the RCP4.5 scenario compared with its reference are estimated to be higher in the eastern U.S. (ranging from 0.6-1.0 $\mu g\ m^{-3}$) than the west (0-0.4 $\mu g\ m^{-3}$) for $PM_{2.5}$, with an average of 0.47 $\mu g\ m^{-3}$ over U.S.; for $O_3$, the total co-benefits are more uniform at 2-5 ppb with U.S. average of 3.55 ppb. Comparing the two mechanisms of co-benefits, we find that reductions of co-emitted air pollutants have a much greater influence on both $PM_{2.5}$ (96% of the total co-benefits) and $O_3$ (89% of the total) than the second co-benefits mechanism via slowing climate change, consistent with West et al. (2013). GHG mitigation from foreign countries contributes more to the U.S. $O_3$ reduction (76% of the total) than that from domestic GHG mitigation only (24%), highlighting the importance of global methane reductions and the intercontinental transport of air pollutants. For $PM_{2.5}$, the benefits of domestic GHG control are greater (74% of total). Since foreign contributions to the co-benefits are comparable to that from the domestic reductions, especially for $O_3$, previous studies that focus on local or regional co-benefits may greatly underestimate the total co-benefits of global GHG reductions. We conclude that the U.S. can gain significantly greater domestic air quality co-benefits by engaging with other nations to control GHGs.



## 1 Introduction

Climate change and air quality are interrelated problems. First, climate change can affect the formation, destruction and transport of major air pollutants, through changes in meteorological variables of temperature, precipitation, air stagnation events, etc. (Weaver et al., 2009; Jacob and Winner, 2009; Fiore et al., 2012, 2015). It can also affect natural emissions (biogenic, dust, fire and lighting) that influence air quality. Second, air pollutants such as particulate matter (PM) and ozone ($O_3$) can change the climate by altering the solar and terrestrial radiation balance through direct and indirect effects (Myhre et al., 2013). Third, the sources of emissions of greenhouse gases (GHGs) and air pollutants are usually shared, particularly through the combustion of fossil fuels, so actions to control one can also influence emissions of the other. Policies to control GHG emissions will therefore not only slow climate change in the future, but will also provide co-benefits of improvements to air quality and consequently to human health (Bell et al., 2008; Nemet et al., 2010).

Recent studies that model future air quality have focused on single or combined changes in future climate and emissions on global and regional air quality, using both global and regional Chemical Transport Models (CTMs) (Weaver et al., 2009; Jacob and Winner, 2009; Fiore et al., 2012). Climate change is likely to decrease background $O_3$ over remote places due to the elevated humidity, and increase $O_3$ over urban and polluted areas, in part because of higher temperature. Jacob and Winner (2009) concluded that future climate change could increase summertime $O_3$ by 1-10 ppb over polluted regions in the U.S. in scenarios from the Special Report on Emission Scenarios (SRES; Nakicenovic and Swart, 2000). In one study, climate change in 2050 under the SRES A1B scenario is projected to increase summertime $O_3$ by 2-5 ppb over large areas in the U.S., comparable to the effect of reduced anthropogenic emissions which reduces $O_3$ by 2-15 ppb, especially in the east (Wu et al., 2008). The overall effect of climate change on PM is less clear, as different components of PM may respond differently to changes in climate variables (Jacob and Winner, 2009; Tai et al., 2010; Fiore et al., 2012, 2015).

Many studies have also estimated the co-benefits of regional or local GHG mitigation on air quality and human health through reductions in co-emitted air pollutants. Cifuentes et al. (2001) found that GHG mitigation through reduced fossil fuel combustion could bring significant local air pollution-related health benefits to some megacities. These health benefits have been estimated in many studies (Bell et al., 2008), and give co-benefits ranging from \$2–196 /tCO$_2$ when monetized, comparable to the costs of GHG reductions (Nemet et al., 2010). A few studies also analyze the co-benefits on future air quality and human health from future regional GHG mitigation scenarios (Thompson et al., 2014; Trail et al., 2015). Thompson et al. (2014) studied the co-benefits of different U.S. climate policies on 2030 domestic air quality, and found that when monetized, the human health benefits due to the improved air quality can offset 26–1050% of the cost of the carbon polices, depending on the policy.

These co-benefits studies may underestimate the total co-benefits as they only consider local or regional climate policies, neglecting benefits outside of the region considered, and benefits within those regions from global GHG mitigation. The total co-benefits of global mitigation are relevant as meaningful GHG mitigation requires participation from at least several of the most highly-emitting nations. We examined the co-benefits of global GHG reductions on both global and regional air





quality and human health, using a global atmospheric model (Model for OZone And Related chemical Tracers, version 4, MOZART-4, hereafter referred to as MZ4) and self-consistent future scenarios (West et al., 2013, referenced hereafter as WEST2013). In addition to evaluating co-benefits through reductions in co-emitted air pollutants, WEST2013 was the first study to quantify co-benefits through a second mechanism: slowing climate change and its effects on air quality. There are

several other innovations of WEST2013: we account for global air pollution transport and long-term influences of methane using the global CTM; we consider realistic scenarios in which air pollutant emissions, demographics, and economic valuation are modeled consistently; and we evaluate chronic mortality influences of fine PM ($PM_{2.5}$, PM with diameter smaller than 2.5 µm) as well as $O_3$. WEST2013 concluded that global GHG mitigation could bring significant air quality improvement for both $PM_{2.5}$ and $O_3$, and avoid 2.2±0.8 million premature deaths globally by 2100 due to the improved air

quality. When monetized, the global average marginal co-benefits of avoided mortality were \$50–380/$tCO_2$, higher than the previous estimates (Nemet et al., 2010). The co-benefits from the first mechanism of reduced co-emitted air pollutants were shown to be much greater than the co-benefits from the second mechanism via slowing climate change.

The WEST2013 study is limited by the coarse resolution of the CTM used (2°×2.5° horizontally). Here we investigate the co-benefits of global GHG mitigation on U.S. air quality at much finer resolution (36km×36km), building on the scenarios in

the global study. WEST2013 simulated co-benefits in 2030, 2050, and 2100, and we choose here to downscale the results in 2050, as climate change influences air quality by 2050 and it is within the timeframe of current decision-making for both climate change and air quality. We use a comprehensive modeling framework in the downscaling process, including a regional climate model to dynamically downscale the global climate to the contiguous United States (CONUS), an emissions processing program to directly process the global anthropogenic emissions to the regional scale, and we create dynamical

boundary conditions (BCs) from the global co-benefits outputs for the regional CTM. We quantify the total co-benefits of global GHG mitigation on U.S. air quality for both $PM_{2.5}$ and $O_3$, and then separate the co-benefits from the two mechanisms analyzed by WEST2013. We also quantify the co-benefits from domestic GHG mitigation versus the co-benefits from those of foreign countries' reductions. We then present the co-benefits from global and domestic GHG mitigation on nine U.S. regions.

With regard to previous studies on the effect of climate change on future air quality (e.g. Jacob and Winner, 2009), our work differs in our reframing of this impact as a co-benefit of slowing climate change from GHG mitigation, and by analyzing that co-benefit through realistic future scenarios, following WEST2013. With regard to previous co-benefits studies that have been conducted on a regional scale (e.g., Thompson et al., 2014), this research differs by embedding the regional co-benefits study in consistent global context, accounting for the effects of changes in global air pollutant emissions and climate change

on U.S. air quality.



## 2 Methodology

Future air quality changes under global and regional GHG mitigation scenarios are simulated using a regional CTM. The scenarios modeled here are built on those of WEST2013, who compared the Representative Concentration Pathway 4.5 (RCP4.5) scenario with its associated reference scenario (REF). Air pollutant emissions in REF are state of the art long-term

emissions projections created by using the Global Change Assessment Model (GCAM) (Thomson et al., 2011). RCP4.5 was developed based on REF by applying a global carbon price to all world regions and all sectors including carbon in terrestrial systems. As discussed by van Vuuren et al. (2011), the air pollutant emissions for the four RCP scenarios were prepared by different groups using different models and assumptions, so they are inconsistent with one another. But by comparing REF with RCP4.5, we use a self-consistent pair of scenarios, where the difference is uniquely attributed to a climate policy.

WEST2013 used both emissions and meteorology from RCP4.5 to simulate future air quality under the RCP4.5 climate policy, and used emissions from REF and meteorology from RCP8.5 to simulate future air quality assuming no climate policy. Since no General Circulation Model (GCM) conducted future climate simulations for the REF scenario, RCP8.5 is used as a proxy for the future climate under REF. The differences between these two scenarios give the total co-benefits for future air quality under climate policy from RCP4.5. Through one extra simulation with emissions from RCP4.5 together

with RCP8.5 meteorology (e45m85 in Table 1), and by comparing with REF and RCP4.5, WEST2013 separated the total co-benefits into the two mechanisms: the co-benefits from reductions in co-emitted air pollutants, and co-benefits from slowing climate change and its influence on air quality.

Here we conduct downscaling processes to provide fine-resolution inputs for the regional CTM. We use the Weather Research and Forecasting model version 3.4.1 (WRF, Skamarock and Klemp, 2008) to downscale the future global climate

from the GCM to the regional scale at a horizontal resolution of 36×36 km for the CONUS. We directly process global anthropogenic emissions to regional scale using the Sparse Matrix Operator Kernel Emissions (SMOKE, v3.5, https://www.cmascenter.org/smoke/) program. The outputs from the global MZ4 simulations of WEST2013 (Table 1) are downscaled to provide initial condition (IC) and dynamic hourly BCs for the regional CTM. The latest version of the Community Multi-scale Air Quality model (CMAQ, v5.0.1, Byun and Schere, 2006) is used as the regional CTM to simulate

air quality changes over the CONUS domain. WEST2013 simulated five consecutive years for each scenario, and used the last four years' average for the data analysis with the first year as a spin-up. Due to the limitations of computational resources, we run CMAQ for 40 months consecutively for each scenario, with the first 4 months as spin-up, and analyze the results as three-year averages.

### 2.1 Regional meteorology

WEST2013 used NOAA Geophysical Fluid Dynamics Laboratory (GFDL) atmospheric model AM3 (Donner et al., 2011; Naik et al., 2013) simulations to provide global meteorology for MZ4. Here we dynamically downscale GFDL AM3, which has a horizontal resolution of 2°×2.5°, to 36×36 km over the CONUS using the WRF model. GFDL AM3 meteorology for





the two RCP scenarios (RCP8.5 and RCP4.5) in 2050 used by WEST2013 is downscaled using a one-way nesting configuration for five consecutive years. WRF is initialized at 0000 Coordinated Universal Time (UTC) 1 January 2048 and run for a 12-month spin-up, then run continuously through 0000 UTC 1 January 2053. A historical period from GFDL AM3 is also downscaled with WRF initialized at 0000UTC 1 January 1999 and run for a 12-month spin-up, then run continuously

through 0000 UTC 1 January 2004. The WRF physics options include the Rapid Radiative Transfer Model for global climate models (Iacono et al., 2008) for longwave and shortwave radiation, WRF single-moment 6-class microphysics scheme (Hong and Lim, 2006), the Grell ensemble convective parameterization scheme (Grell and Devenyi, 2002), the Yonsei University planetary boundary layer scheme (Hong et al., 2006), and the Noah land surface model (Chen and Dudhia, 2001). The WRF configuration also applies spectral nudging. Otte et al. (2012) and Bowden et al. (2012, 2013) demonstrated that

using nudging in WRF improves the overall accuracy of the simulated climate over the CONUS at 36-km and does not squelch extremes in temperature and precipitation. In particular, spectral nudging affects the model solution through a nonphysical term in the prognostic equations based on the difference between the spectral decomposition of the model solution and the reference analysis. Spectral nudging is used to constrain WRF toward synoptic-scale wavelengths resolved by GFDL AM3 exceeding 1200 km. Nudging is applied equally to potential temperature, wind, and geopotential with a

nudging coefficient of $1.0 \times 10^{-4}$, which is equivalent to a time scale of 2.8 hours. The downscaled meteorology from WRF is used to provide meteorological inputs to CMAQ. Hourly WRF outputs are processed using Meteorology-Chemistry Interface Processor (MCIP v4.1; Otte and Pleim, 2010) to provide meteorological inputs for CMAQ.

Comparing the downscaling results between WRF with the GFDL AM3 simulation for three-year averages of the 2-m temperature (we present three-year averages instead of four to be consistent with CMAQ outputs below), we see that the

large-scale spatial patterns for temperature are similar (Fig. S1). However, the downscaling clearly improves the resolved features related to topography and provides a different realization of average regional climate throughout the CONUS. Comparing WRF future projected change centered on 2050 with 2000, we see that the three-year average of 2-m temperature generally increases over the entire U.S. for both RCP8.5 and RCP4.5 (Fig. S2-S3). Temperature increases are largest for extreme northeastern latitudes, the Southeast and Southwest U.S. in both scenarios, with U.S. average warming of 3.05°C

and 2.59°C for RCP8.5 and RCP4.5, respectively. Additionally, precipitation is projected to increase over most of the U.S. in both scenarios with U.S. average increases of 8.16 and 7.63 mm day$^{-1}$ in RCP8.5 and RCP4.5. Comparing the changes between scenarios (RCP8.5 minus RCP4.5), Fig. 1 illustrates that temperature increases are smaller in RCP4.5 throughout the CONUS, except in the Northwest. The precipitation difference between scenarios has a larger spatial variability than the 2-m temperature. However, the only region where the regional climate is warmer and drier in RCP4.5 is in the Northwest

U.S. Ignoring other influences of climate change, increases in precipitation would be expected to increase PM wet scavenging, and decrease PM concentration.



## 2.2 Regional emissions

Similar studies in the past have typically chosen to run SMOKE with the present-day U.S. National Emission Inventory (NEI), and then scale the SMOKE outputs into future years, using the mass ratio of projected future to present-day emissions from global inventories (e.g., Hogrefe et al., 2004; Nolte et al., 2008; Avise et al., 2009; Chen et al., 2009; Gao et al., 2013).

Instead, we use SMOKE to directly process the global emissions in 2000 and in 2050 from REF and RCP4.5 to provide temporally- and spatially-resolved CMAQ emission input files. We first regrid the global emissions datasets at $0.5° \times 0.5°$ into finer resolution (36km×36km), and then take advantage of the temporal and speciation profiles inside SMOKE to assign temporal variations and re-speciate the PM and VOCs species. By doing this, we account better for the spatial distribution changes of future emissions projected in the RCPs (Figs. S4-S10), whereas the traditional method only considers changes in

the magnitude of air pollutants in the future, assuming a constant spatial and sectoral distribution.

In addition, the RCP datasets report only elemental carbon (EC) and organic carbon (OC), but ignore emissions of other primary PM species. Here we back-calculate the total $PM_{2.5}$ and PM coarse (PMC) primary emissions for all sectors from the reported EC and OC. We first derive the emission fractions of EC and OC in each sector by cross-comparing the definitions of the sectors in IPCC, the Source Clarification Codes (SCC) in the speciation cross-reference file

(http://www.airqualitymodeling.org/cmaqwiki/index.php?title=CMAQv5.0_GSREF_example, accessed 5 September 2013), and the EPA PM speciation profile file built into SMOKE (http://www.airqualitymodeling.org/cmaqwiki/index.php?title=CMAQv5.0_GSPRO_Example, accessed 5 September 2013) (Table S1). If multiple sources are included in one IPCC sector (e.g., energy and industries in Table S1), we use the mass ratio from the source that contributes the largest fraction by referring to previous studies (Reff et al., 2009; Xing et al., 2013).

Then we calculate the total $PM_{2.5}$ and PMC in each grid cell by dividing the reported EC and OC by their emission fractions individually, and average these two. By doing this, we increase the total $PM_{2.5}$ emissions of the RCPs by incorporating the inorganic components of primary PM, such as sulfate and nitrate. We check these results by comparing the total 2000 $PM_{2.5}$ emissions of 4.14 Tg $yr^{-1}$ in this study (Table 2) with other studies, finding that it is comparable to the total of 4.69 Tg $yr^{-1}$ in 2001 from the U.S. NEI (http://www.epa.gov/ttnchie1/trends/, accessed 5 October 2013). Our calculated $PM_{2.5}$ emission is

also lower than the estimated 5.53 Tg $yr^{-1}$ in 2000 by Xing et al. (2013), which used an activity data based approach to develop consistent temporally-resolved emissions from 1999 to 2010.

In Table 2, we list the U.S. anthropogenic emissions for major air pollutants in 2000 and 2050 from REF and RCP4.5. Significant decreases are seen for most pollutants from 2000 to 2050 for both REF and RCP4.5, except for $NH_3$ which is projected to increase due to agricultural activity (van Vuuren et al., 2011). Comparing RCP4.5 and REF, emissions of $PM_{2.5}$

and $O_3$ precursors also decrease, including EC (7.59%) and OC (6.17%), with $NO_x$ and NMVOC decreasing by more than 10%. $SO_2$ has the largest relative decreases between RCP4.5 and REF in 2050 (28.78%). Large spatial variations in emissions reductions are also seen over the U.S., with the largest reductions seen on the east and west urban areas of U.S. for most air pollutants and smaller reductions in the Great Plains (Figs. S4-S10).



Biogenic emissions are estimated using the Biogenic Emission Inventory System (BEIS v3.14), which responds to the changing climate for different scenarios. It is configured to run on-line in CMAQ, and calculates the emissions of 35 chemical species including 14 monoterpenes and 1 sesquiterpene. We assume that land use and land cover will stay constant in the future for the purpose of estimating biogenic emissions. The on-line option of lightning is also turned on to calculate

the $NO_x$ emissions by estimating the number of lightning flashes based on the modeled convective precipitation, which also changes with climate. We prepare the ocean/land mask for the domain to calculate sea salt emissions which can be significant in coastal environments (Kelly et al., 2010). We also use the BEIS on-line calculation for natural soil $NO_x$ emissions.

## 2.3 Regional air quality model and dynamical chemical BCs

The latest CMAQ model (https://www.cmascenter.org/cmaq/index.cfm, accessed 15 June 2012) is used to perform the regional air quality simulations with the CB05 chemical mechanism and updated toluene reactions. The model incorporates the newest aerosol module (AE6), including features of new PM speciation (Reff et al., 2009), oxidative aging of primary organic carbon (Simon and Bhave, 2012), and an updated treatment and tracking of crustal species (e.g., $Ca^{2+}$, $K^+$, $Mg^{2+}$) and trace metals (e.g., Fe, Mn) (Fountoukis and Nenes, 2007). Several other enhancements in v5.0 of CMAQ were discussed by

Appel et al. (2013) and Nolte et al. (2015), and there are no significant changes for the aerosol module between v5.0 and v5.0.1

(http://www.airqualitymodeling.org/cmaqwiki/index.php?title=CMAQ_version_5.0.1_%28July_2012_release%29_Technical_Documentation, accessed 15 August 2012). The model is configured with 34 vertical layers, with the lowest level being 34 m high, to the highest level at 50 hPa. The horizontal resolution is 36 km by 36 km for the CONUS domain. $PM_{2.5}$ is

calculated from the CMAQ output as the sum of the species EC, OC, secondary organic aerosol (SOA), non-carbon organic matter (NCOM), nitrate ($NO_3^-$), sulfate ($SO_4^{2-}$), ammonium ($NH_4^+$), sodium ($Na^+$), chloride ($Cl^-$), eight crustal and trace metal species, and other unspeciated fine PM (OTHER).

The dynamical BCs for this study are provided by the global MZ4 simulations of WEST2013. The hourly boundary values from MZ4 are horizontally interpolated from coarser resolution to the regional finer resolution, and also vertically

interpolated as MZ4 and CMAQ have different vertical layers. Chemical species are mapped between MZ4 and CMAQ v5.0.1, due to the different chemical mechanisms used by these two models, following the descriptions of Emmons et al. (2010) and ENVIRON (http://www.camx.com/download/support-software.aspx, accessed 19 September 2013). For the chemical species in CMAQ that do not exist in MZ4, values are set to defaults as suggested by the CMAQ website.

## 2.4 Scenarios

We simulate scenarios in CMAQ comparable to WEST2013, except that we carry out one extra scenario to quantify the co-benefits from domestic versus foreign GHG mitigation (Table 1). S_2000 is conducted to evaluate CMAQ model performance and to compare with future scenarios. For this study, we run four scenarios in 2050. The differences between



S_RCP45 and S_REF are the total co-benefits on U.S. air quality from global GHG mitigation. The emission benefit from the first mechanism is calculated as the difference between S_Emis and S_REF, and the meteorology benefit is calculated as S_RCP45 minus S_Emis. By comparing S_Dom (applying GHG mitigation from RCP4.5 scenario in the U.S. only) with S_REF, and S_RCP45 with S_Dom, we quantify the co-benefits from domestic and foreign GHG mitigation. In estimating the co-benefits of domestic reductions, we account for the influences of global climate change as a foreign influence (as most GHG emissions are global), assuming that U.S. air pollutant emissions have small effects on global or regional climate, such as through aerosol forcing. In each scenario, we fix global methane at concentrations given by the RCPs (Table 1), and account for methane changes as a foreign influence, neglecting the fraction of global methane emissions that are from the U.S. All scenarios are set up as continuous runs, with S_2000 running from September, 2000 to December, 2003, with the first four months in 2000 as spin-up. The future scenarios are run from September, 2049 to December, 2052 with the months in 2049 as spin-up. Results are presented as the average of three years.

## 3 Results

### 3.1 CMAQ model evaluation

The CMAQ model has been broadly used to study regional future air quality (Hogrefe et al., 2004; Tagaris et al., 2007; Nolte et al., 2008; Lam et al., 2011; Gao et al., 2013) and has been evaluated in many applications (Appel et al., 2010, 2011, 2013; Nolte et al., 2015). Here we evaluate the CMAQ v5.0.1 performance by comparing the model outputs from S_2000 with observations in 2000 from the Interagency Monitoring of PROtected Visual Environments (IMPROVE; http://vista.cira.colostate.edu/improve/, accessed 9 May 2014), the Chemical Speciation Network (CSN; previously known as STN, http://www.epa.gov/ttn/amtic/speciepg.html, accessed 9 May 2014), and the Clean Air Status and Trends Network (CASTNET; http://epa.gov/castnet/javaweb/index.html, accessed 9 May 2014) for total $PM_{2.5}$ and its components, and the EPA Air Quality System (AQS; http://www.epa.gov/ttn/airs/airsaqs/detaildata/downloadaqsdata.htm, accessed 9 May 2014) for $O_3$. We pair the model outputs with observations in space and time, and calculate four groups of statistics to evaluate model performance: Median Bias (MdnB, µg m$^{-3}$ for $PM_{2.5}$ and ppb for $O_3$), Normalized Median Bias (NMdnB, %), Median Error (MdnE, µg m$^{-3}$ and ppb) and Normalized Median Error (NMdnE, %) (Supplementary material). Median metrics are used here instead of the mean, as for data with non-normal distributions (i.e., PM species) the median gives a better representation of the central tendency of the data (USEPA 2007). For $O_3$ evaluation, we use both the maximum daily 1-hour (1hr_$O_3$) and Maximum Daily 8-hour Average (MDA8), and also calculate these metrics with a cutoff value of 40 ppb for the observed $O_3$ to evaluate the model's reliability in predicting ozone values relevant for the NAAQS (USEPA, 2007). Model performance is not expected to be perfect as meteorology does not correspond with actual year 2000 meteorology, and emissions are derived from global datasets rather than specific emissions for the U.S.

For total $PM_{2.5}$, overall model performance is good and the NMdnE for IMPROVE and CSN are less than 50%, with slight differences in performance (Table 3). CMAQ underestimates $PM_{2.5}$ in these two networks and also its components in all





three networks (Table S2), except that it overestimates $SO_4^{2-}$ compared with IMPROVE, and $NH_3^+$ with CSN. Compared with other components, OC and EC are not well predicted, with higher NMdnB, -63.55% and -37.00% in IMPROVE (OC and EC are not measured in the other two networks). In simulating $PM_{2.5}$ and its species, model performance is better in winter than in summer (not shown here). The model overestimates surface $O_3$ as indicated by the positive MdnB (ppb) and

NMdnB (%). The NMdnE for the 1hr-$O_3$ (MDA8-$O_3$) declines from 27.60% (33.35%) to 17.36% (16.95%) after we apply the cutoff value of 40 ppb. The overprediction is slightly higher for 1hr-$O_3$ than for MDA8-$O_3$, however this difference becomes smaller when we consider the cutoff values.

### 3.2 Air quality changes in 2050

Here we show the seasonal and spatial patterns of future air quality changes centered in 2050 relative to 2000 from REF and

RCP4.5 (Figs. S11 to S14). The three-year seasonal average of $PM_{2.5}$ over the entire U.S. decrease in 2050 in both S_REF and S_RCP45 compared with S_2000, especially in the Eastern U.S. and California (CA). The seasonal decreases are largest in winter, with U.S. averages in S_REF (S_RCP45) of 4.42 (4.88) µg m$^{-3}$, and lowest in the summer of 1.55 (2.00) µg m$^{-3}$, with annual average of 2.76 (3.23) µg m$^{-3}$. The three-year seasonal average of $O_3$ decrease significantly in summer in both the east and west coast, with U.S. average of 6.31 (9.50) ppb in S_REF (S_RCP45). $O_3$ increases over the Northeast and

West U.S. in winter in both S_REF and S_RCP45, caused by the weakened $NO_x$ titration as a result of the large $NO_x$ decrease in the two scenarios (Table 2), as also reported in other studies (Gao et al., 2013; Fiore et al., 2015). The magnitude of the decreases between S_REF and S_2000 is lower than that between S_RCP45 and S_2000, as the REF scenario did not apply a GHG mitigation policy, and thus has less emission reductions.

We then compare these air quality changes in 2050 with the MZ4 simulations of WEST2013 for both S_REF (Fig. S15) and

S_RCP45 (Fig. 2), and for S_RCP45 with the ensemble model means from the Atmospheric Chemistry and Climate Model Intercomparison Project (ACCMIP, Lamarque et al., 2013) following Fiore et al. (2012), as no ACCMIP models simulated REF in 2050. For the U.S. annual average $PM_{2.5}$, the decrease in 2050 for S_RCP45 relative to 2000 in this study (3.23 µg m$^{-3}$) is modestly higher than both the results from MZ4 and the ACCMIP ensemble mean, but within the range of ACCMIP models when $PM_{2.5}$ is calculated as a sum of species. The future $O_3$ changes in our study (5.20 ppb) are clearly in

the range of ACCMIP results, and nearly identical to MZ4 (5.13 ppb). Comparisons of the air quality changes in 2050 for S_REF relative to 2000 between CMAQ and MZ4 are similar, except that the magnitudes of the changes are smaller than those for S_RCP45 (Fig. S15).

### 3.3 Total co-benefits for U.S. air quality from global GHG mitigation

Projected three-year average $PM_{2.5}$ concentrations in 2050 in both scenarios (S_REF and S_RCP45) are higher in the Eastern

U.S. and the west coast of CA, and lower in the Western U.S. (Fig.3). The total co-benefits for U.S. air quality (S_RCP45 minus S_REF) show notable decreases of major air pollutants in 2050. The total co-benefits for $PM_{2.5}$ over the U.S. show a significant spatial gradient over the U.S. domain, greatest in the eastern U.S., especially urban areas, as well as CA, ranging





from 0.4 to 1.0 µg m$^{-3}$, and least in the Rocky Mountains and Northwest with values below 0.4 µg m$^{-3}$. The total co-benefits for PM$_{2.5}$ averaged over the U.S. is 0.47 µg m$^{-3}$, with the largest contribution from organic matter (OM, including primary OC, SOA and NCOM), accounting for the 45% of the total (0.21 µg m$^{-3}$), followed by sulfate (0.11 µg m$^{-3}$) and ammonia (0.05 µg m$^{-3}$) (Fig. S16). The total co-benefits are highest in fall, with U.S. domain average of 0.55 µg m$^{-3}$, and lowest in

spring (0.41 µg m$^{-3}$) (Fig. 4). Notice that the region with greatest co-benefits shifts from Central areas in winter and spring to the East in summer and fall, with the largest component of OM also shifting from primary OC to SOA (Fig. S17).

Future O$_3$ is presented here as the ozone-season average (from May to October) of MDA8. In general, 2050 O$_3$ concentrations in S_REF and S_RCP45 are projected to be high in the Southern U.S., especially over the coastal areas, and higher in the West than the East (Fig. 5). The total co-benefits for O$_3$ are fairly uniformly significant over the entire U.S.

domain, slightly higher in the Northeast and Northwest, and range from 2-5 ppb with a domain average of 3.55 ppb, unlike PM$_{2.5}$ which is higher over urban regions. The uniformity of the total O$_3$ co-benefits suggests that they are strongly influenced by global O$_3$ reductions.

The total co-benefit for PM$_{2.5}$ from this study (0.47 µg m$^{-3}$ over U.S.) is lower than WEST2013 (area-weighted three-year averages of 0.72 µg m$^{-3}$ over U.S.), especially over the Northwest and Central of U.S. (Fig. S18). Analyzing the components

of PM$_{2.5}$, we find that this difference is mainly caused by OM, with a U.S. annual average of 0.40 µg m$^{-3}$ in WEST2013 and 0.21 µg m$^{-3}$ in this study (Fig. S19). For other components (EC, SO$_4^{2-}$, NO$_3^-$ as reported in MZ4 of WEST2013), the CMAQ results are slightly lower than WEST2013 but share a similar spatial pattern (Figs. S20-S22). We expect that the total co-benefits of PM$_{2.5}$ in this study might be higher than WEST2013, as we account for inorganic primary PM emissions in SMOKE. A possible explanation may be that different chemical mechanisms and deposition processes are adopted for

organic aerosols in MZ4 and CMAQ, which may make a shorter atmospheric lifetime for PM in CMAQ than that in MZ4. The differences of the meteorology (e.g., the precipitation and temperature) between the downscaled WRF and the GFDL could also contribute to this difference. Total co-benefit of O$_3$ from this study (3.55 ppb over U.S.) is comparable to WEST2013 (3.71 ppb) in both the magnitude and spatial distribution (Fig. S23).

### 3.4 Co-benefits from the two mechanisms

We quantify the co-benefits of global GHG mitigation on PM$_{2.5}$ and O$_3$ through the two mechanisms: reduced co-emitted air pollutants (S_Emis—S_REF) and slowing climate change and its effect on air quality (S_RCP45—S_Emis). The reduction of co-emitted air pollutants has a much greater effect than slowing climate change for PM$_{2.5}$, accounting for 96% of the U.S. average PM$_{2.5}$ decrease. The emission benefit for PM$_{2.5}$ over the U.S. domain is 0.45 µg m$^{-3}$, greatest near urban areas where emissions are reduced (Fig. 6), with the largest contribution from OM (0.172 µg m$^{-3}$ over the U.S.), followed by sulfate

(0.107 µg m$^{-3}$) and ammonia (0.048 µg m$^{-3}$). Slowing climate change only accounts for 4% of the U.S. average total PM$_{2.5}$ decreases (0.02 µg m$^{-3}$). It also has different signs of effect over the U.S., reducing PM$_{2.5}$ in the Southern U.S. but increasing in the North.





For $O_3$, the emission benefit is also larger than the climate benefit, accounting for 89% of the total $O_3$ decreases averaged over the U.S. The emission benefit for $O_3$ over the U.S. domain is 3.16 ppb, and much more uniform over the U.S., slightly higher over Northeast and Northwest. Slowing climate change accounts for 0.39 ppb $O_3$ decreases, 11% of the total and mainly in the Great Plains and the East, where temperatures are cooler under RCP4.5 compared with RCP8.5 (Fig. 1). The dominance of the emission co-benefit over the climate co-benefit for both $PM_{2.5}$ and $O_3$ is consistent with WEST2013.

## 3.5 Co-benefits from domestic and foreign GHG mitigation

We also investigate the co-benefits from domestic GHG mitigation by comparing S_Dom with S_REF, versus foreign GHG reductions by comparing S_RCP45 with S_Dom (Fig. 7). For $PM_{2.5}$, domestic GHG mitigation accounts for 74% (0.35 µg m$^{-3}$) of the total $PM_{2.5}$ decrease over the whole U.S., with the greatest effect over the East and CA, where emissions of $PM_{2.5}$ and its precursors are greatly reduced (Figs. S3-S9). The benefits from foreign GHG reductions on the U.S. $PM_{2.5}$ change are only obvious in the Southern U.S., influenced by emission reductions in Mexico and global climate change. We conclude that domestic GHG mitigation has a greater influence on U.S. $PM_{2.5}$ than reductions in foreign countries, but that foreign reductions also make a noticeable contribution, accounting for 26% of total $PM_{2.5}$ decreases over the U.S., and a greater fraction in the Southern U.S.

For $O_3$, foreign countries' GHG mitigation has a much larger influence on the U.S., accounting for 76% (2.69 ppb) of the total $O_3$ decrease, compared with 24% from domestic GHG mitigation (Fig. 7). The U.S. experiences greater $O_3$ decreases in the North than the South, which is likely influenced in part by the air quality improvement in Western Canada as a result of slowing deforestation due to the climate policy in RCP4.5 (West et al., 2013). This large influence of foreign reductions for $O_3$ highlights the importance of global methane reductions in RCP4.5 and global emission reductions, particularly in Asia and intercontinental transport.

## 3.6 Regional co-benefits and variability

We then quantify the co-benefits over nine U.S. climate regions defined by the National Oceanic and Atmospheric Administration (Fig. S24), and their domestic and foreign components. The Central, Southeast, Northeast and South regions have the largest total co-benefits for $PM_{2.5}$ (regional annual means of 0.78, 0.75, 0.62 and 0.62 µg m$^{-3}$), and the Northwest has the lowest total co-benefits (0.16 µg m$^{-3}$) (Fig. 8). Domestic GHG mitigation has the largest effect over these same regions and lowest effects over Northwest and West North Central, with means of 0.13 µg m$^{-3}$. Foreign co-benefits are greatest over the South, Southwest, Central and Southeast, and lowest over Northwest (Table S3). As a fraction of the total co-benefits, the domestic co-benefit is highest in the Northeast, East North Central and Central accounting for more than 80% of the total, while foreign co-benefits are highest over Southwest, South and West North Central, accounting for about 40% of the total.

For $O_3$, the Northeast, East North Central, and Northwest have the highest total co-benefits, (regional means of 4.61, 4.25, 4.15 ppb; Fig. 9 and Table S3), although the total co-benefits for $O_3$ are fairly uniform over the U.S (Fig. 5). The Southeast




has the lowest total co-benefits, with 2.67 ppb for the regional mean. Domestic co-benefits are higher over the Central, Northeast and Southeast, with regional means of 1.25, 1.16 and 1.14 ppb, and lowest over Northwest (0.4 ppb). In general, foreign mitigation contributes more in the west than the east, most likely influenced by intercontinental transport from Asia. It is highest in the Northwest, West North Central and Northeast, with regional means of 3.75, 3.45 and 3.45 ppb. The fraction of co-benefits from foreign mitigation is larger than 60% in most regions, highest over the Northwest (90%), and lowest over the Southeast (57%).

We also evaluate the variability in co-benefits for the three years simulated (Table S3). Over the U.S., the coefficient of variation (CV) for the total co-benefits for $PM_{2.5}$ (7%) is much lower than that of the total co-benefits for $O_3$ (37%), which is controlled by the intercontinental transport and global $CH_4$. The Southeast has the highest CV (29%) for the total co-benefits of $PM_{2.5}$, while other regions are lower than 15%, lowest in the East North Central and Northeast (3%). Southwest and South have the highest CV (70%, 69%) for the total co-benefits of $O_3$, and lowest in Northwest (21%). For regions with higher variability, longer simulations would be desirable to better quantify the annual average co-benefits.

## 4 Discussion

The co-benefits we present here are specific to the reference (REF) and mitigation (RCP4.5) scenarios we choose, and results would differ for other baseline and mitigation scenarios. The estimated co-benefits also depend on participation of many nations in the mitigation policies, and delaying participation will likely change the co-benefits.

The total co-benefits for $O_3$ when downscaled are comparable to the global study in both magnitude and spatial pattern, but the downscaled simulations capture some local features better than the global model, such the effects of topography and urban areas. For $PM_{2.5}$, significant differences are seen from the downscaling due to the fine resolution and different chemical mechanisms between the global and the regional model. The resolution we are using for this study (36km by 36 km) is fine enough for us to analyze the co-benefits at a state level, but insufficient to fully resolve urban areas. Finer resolution simulations (such as 12 km by 12 km) with CMAQ or other CTMs can be carried out to better quantify the co-benefits over urban areas.

For this study, uncertainties and errors may exist under the assumptions and choices we make for each model. For example, the co-benefits of $PM_{2.5}$ have large contributions from OC and SOA over the Central and East U.S. (Fig. 4, Fig. S16). However, our model evaluations show that CMAQ greatly underestimates the OC concentration compared with surface observations. New gas-phase and aqueous-phase oxidation pathways for SOA formation are found to play significant roles in producing organic aerosols (Lin et al., 2014; Pye and Pouliot, 2012; Pye et al., 2013), which are missing in the CMAQ version used in this study. We use BEIS model to estimate the biogenic VOC (BVOC) emissions, but studies have shown that the BVOCs from the Model of Emissions of Gases and Aerosols from Nature (MEGAN) are higher than those from BEIS by a factor 2 (Pouliot, 2008; Pouliot and Pierce, 2009), which highlights the uncertainty in representing these emissions and simulating both $PM_{2.5}$ and $O_3$ (Hogrefe et al., 2011).



We assume constant land use in the GCM, WRF and CMAQ when simulating the global and regional climate and estimating the biogenic emissions, which could introduce errors in our results (Unger, 2014; Heald and Spracklen, 2015). When we process the global anthropogenic emissions with SMOKE, we back-calculate the total $PM_{2.5}$ and PMC from OC and BC, which introduces inorganic PM emissions and may make our results for co-benefits of $PM_{2.5}$ higher. By doing this, we account for missing emissions but also increase the total uncertainties in the emission inventory. Spectral nudging is adopted in this study to restrain WRF from drifting from the GCM, which has been shown to be better for some meteorological variables, but spectral nudging better for others (Bowden et al., 2012, 2013; Liu et al., 2012; Otte et al., 2012). Moreover, only one model is used at each step during downscaling, and ensemble model means can be used to reduce the single model's variability. Simulations are based on three-year averages, due to computational limitations, but these three years may reflect meteorological variability and not only climate change. This uncertainty may be greater for the total co-benefits of $O_3$, for which we see greater year-to-year variations than for $PM_{2.5}$. CMAQ simulations could be performed over more years to reduce the influence of the climate variability. In separating domestic and foreign co-benefits, we assume that global and regional climate will be controlled by foreign GHGs emissions, and not influenced by GHG mitigation in the U.S., which may also introduce errors into our results. We similarly attribute the global methane change as a foreign influence, as U.S. methane emissions are a small fraction of the global.

## 5 Conclusions

Climate polices to control GHG emissions will not only have the benefit of slowing climate change, but can also have co-benefits of improved air quality. Previous co-benefits studies focus mostly on local or regional GHG reductions. As a result, these studies omit air quality benefits outside of the domain considered, and neglect benefits from global GHG mitigation. In this study we adopt a systematic approach to quantify the co-benefits from both the global and regional GHG mitigation on regional air quality over U.S. at fine resolution in 2050, building on the global co-benefits study from West et al. (2013). The co-benefits of global GHG mitigation on U.S. air quality are discussed through two mechanisms: reduced co-emitted air pollutants and slowing climate change and its influence on air quality. We also quantify the co-benefits from domestic GHG mitigation versus foreign countries' reduction.

We find that there are significant benefits for both $PM_{2.5}$ and $O_3$ over U.S. by 2050 from the global GHG mitigation in RCP4.5. The total co-benefits for $PM_{2.5}$ are higher in the east than the west, with an average of 0.47 µg m$^{-3}$ over U.S. For $O_3$, the total co-benefits are fairly uniform across the U.S. at 2-5 ppb, with U.S. average of 3.55 ppb. The co-benefits from reductions of co-emitted air pollutants have a greater influence on both $PM_{2.5}$ (accounting for 96% of total decreases) and $O_3$ (89% of the total decreases) than the second mechanism via slowing climate change, consistent with West et al. (2013).

Foreign countries' GHG reductions have a much greater influence on the U.S. $O_3$ reduction (76% of the total), compared with that from domestic GHG mitigation only (24%), highlighting the importance of global methane reductions and the intercontinental transport of air pollutants. For $PM_{2.5}$, the benefits of foreign GHG control are less than domestic, but still a



considerable portion of the total (26%). We conclude that the U.S. can gain significantly greater domestic air quality co-benefits by engaging with other nations for GHG control to combat climate change, especially for $O_3$. This also applies to other nations which can be expected to have ancillary air quality benefits from foreign countries' GHG mitigation. We also conclude that previous studies that estimate co-benefits for one nation or region (e.g., Thomson et al., 2014), may

5   significantly underestimate the full co-benefits when many countries reduce GHGs together, particularly for $O_3$.

**Acknowledgements**

This publication was financially supported by the US Environmental Protection Agency STAR grant #834285, and the National Institute of Environmental Health Sciences grant #1 R21 ES022600-01. Its contents are solely the responsibility of

10   the grantee and do not necessarily represent the official views of the USEPA or other funding sources. USEPA and other funding sources do not endorse the purchase of any commercial products or services mentioned in the publication.



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





Table 1. List of CMAQv5.0.1 simulations in this study. Hourly BCs are from the MOZART-4 (MZ4) simulations of WEST2013. We fix the methane ($CH_4$) background concentrations in CMAQ consistent with the RCP scenarios and WEST2013.

| Years | Scenario | Emissions | Meteorology | BCs | $CH_4$ |
|---|---|---|---|---|---|
| 2000 | S_2000 | 2000 | 2000 | MZ4 2000 | 1766 ppbv |
| 2050 | S_REF | REF | RCP8.5 | MZ4 REF | 2267 ppbv |
| | S_RCP45 | RCP4.5 | RCP4.5 | MZ4 RCP4.5 | 1833 ppbv |
| | S_Emis | RCP4.5 | RCP8.5 | MZ4 e45m85[b] | 1833 ppbv |
| | S_Dom | RCP4.5 for U.S., REF for Can, Mex[a] | RCP8.5 | MZ4 REF | 2267 ppbv |

[a]the part of Canada and Mexico in the domain.

[b]global simulation using RCP4.5 emissions together with RCP8.5 meteorology in 2050.





Table 2. Anthropogenic emissions in the U.S. for major air pollutants in 2000 and 2050 from REF and RCP4.5 (Tg yr$^{-1}$), and the relative differences (Relative Diff) between RCP4.5 and REF in 2050 ((RCP4.5 - REF)/REF×100).

|  | **2000** | **2050 REF** | **2050 RCP4.5** | **Relative Diff (%)** |
|---|---|---|---|---|
| $SO_2$ | 14.84 | 2.46 | 1.75 | -28.78 |
| $NH_3$ | 3.34 | 4.56 | 4.30 | -5.56 |
| $NO_x$ | 19.57 | 4.40 | 3.92 | -10.93 |
| CO | 92.74 | 11.42 | 11.25 | -1.48 |
| NMVOC | 15.23 | 8.07 | 7.16 | -11.21 |
| EC | 0.42 | 0.22 | 0.21 | -7.59 |
| OC | 0.71 | 0.35 | 0.33 | -6.17 |
| $PM_{2.5}$[1] | 4.14 | 1.87 | 1.57 | -15.80 |
| PMC[2] | 11.02 | 5.50 | 4.63 | -15.80 |

[1,2]$PM_{2.5}$ & PMC are the total emissions back-calculated based on the EC & OC.





Table 3. Evaluation of the S_2000 simulation (average of three years modeled) with surface observations in 2000 for $PM_{2.5}$ ($\mu g\ m^{-3}$) and $O_3$ (ppb).

|  | Pollutants | MdnB | NMdnB (%) | MdnE | NMdnE(%) |
|---|---|---|---|---|---|
| IMPROVE | $PM_{2.5}$ | -0.89 | -23.31 | 1.88 | 49.46 |
| CSN | $PM_{2.5}$ | -2.85 | -27.44 | 4.29 | 41.30 |
| AQS | $1hr\_O_3$ | 8.97 | 18.40 | 13.25 | 27.60 |
| AQS | $1hr\_O_3\_40$[a] | 2.79 | 4.76 | 9.89 | 17.36 |
| AQS | $MDA8\_O_3$ | 11.87 | 28.01 | 14.13 | 33.35 |
| AQS | $MDA8\_O_3\_40$[a] | 3.95 | 7.37 | 9.09 | 16.95 |

[a]$1hr\_O_3\_40$ and $MDA8\_O_3\_40$: Observations below 40 ppb are excluded from the comparison.



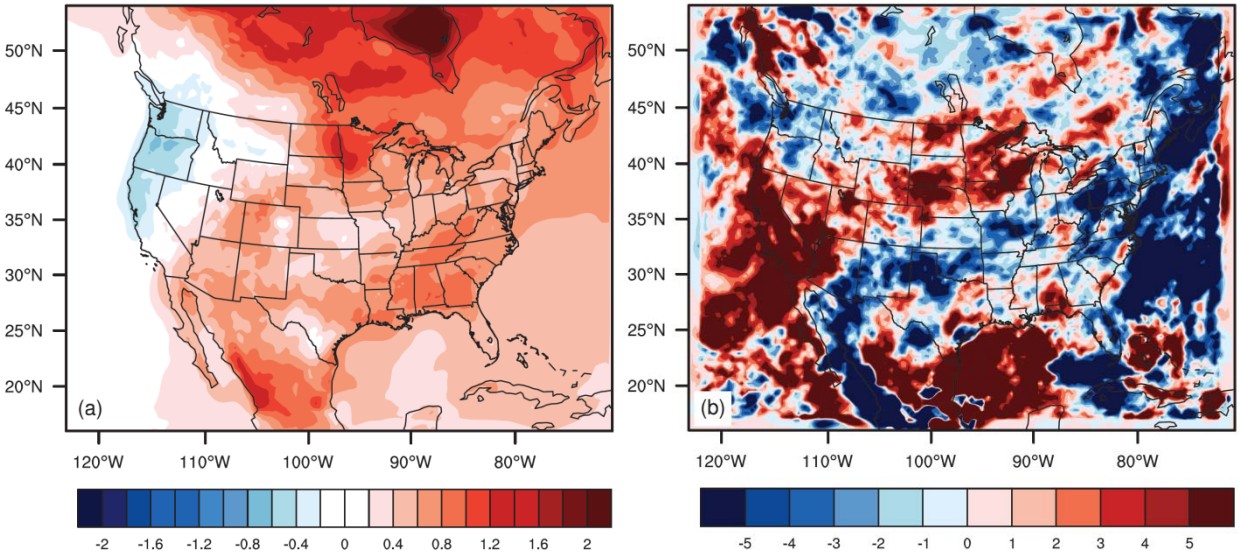

Fig. 1. Changes in (a) 2-m temperature (°C) and (b) precipitation (mm day$^{-1}$) centered on 2050 between RCP8.5 and RCP4.5 (RCP8.5—RCP4.5).




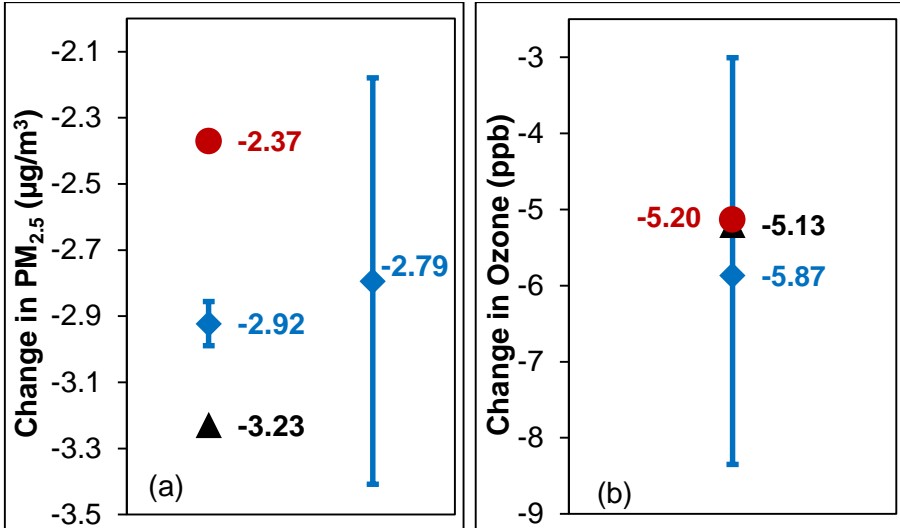

Fig. 2. Comparison of annual U.S. average concentration changes for RCP4.5 in 2050 relative to 2000, for this study (black triangle), MZ4 from WEST2013 (red circle), and the ensemble mean (blue diamond) and multi-model range from ACCMIP (blue lines), for (a) $PM_{2.5}$, and (b) $O_3$. In panel a, the total $PM_{2.5}$ reported by the ACCMIP models is shown on the left, and the $PM_{2.5}$ estimated as a sum of species $BC+OA+SOA+SO_4+NO_3+NH_4+0.25*SeaSalt+0.1*Dust$ following Fiore et al. (2012) and Silva et al. (2013) shown on the right. Values shown are the average of three years for CMAQ and MZ4, and 5 to 10 years for ACCMIP for three models (LMDzORINCA, GFDL-AM3 and GISS-E2-R) that report $O_3$ and two models (GFDL-AM3 and GISS-E2-R) that report $PM_{2.5}$.



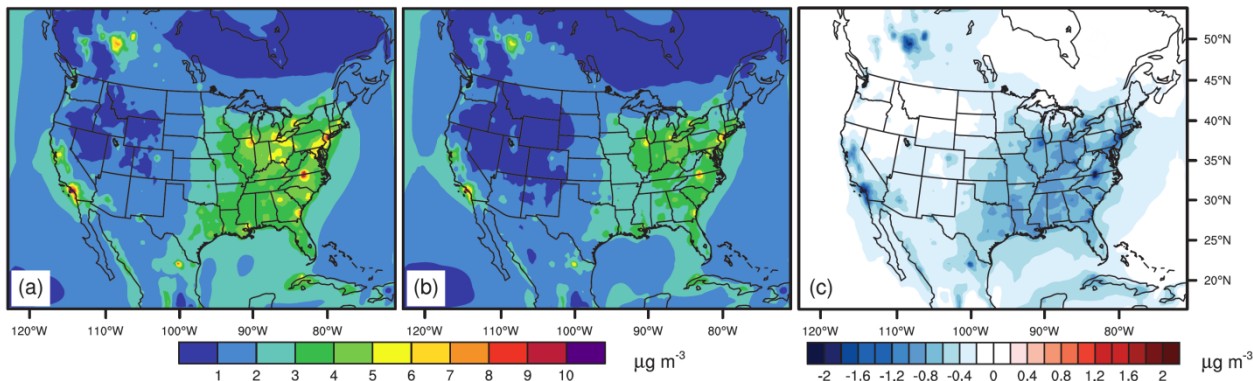

Fig. 3. The three-year average PM$_{2.5}$ (µg m$^{-3}$) distributions in 2050 from (a) S_REF, (b) S_RCP45, and (c) the total co-benefits (shown as the difference between S_RCP45 and S_REF). Blue colors in panel (c) indicate an air quality improvement.





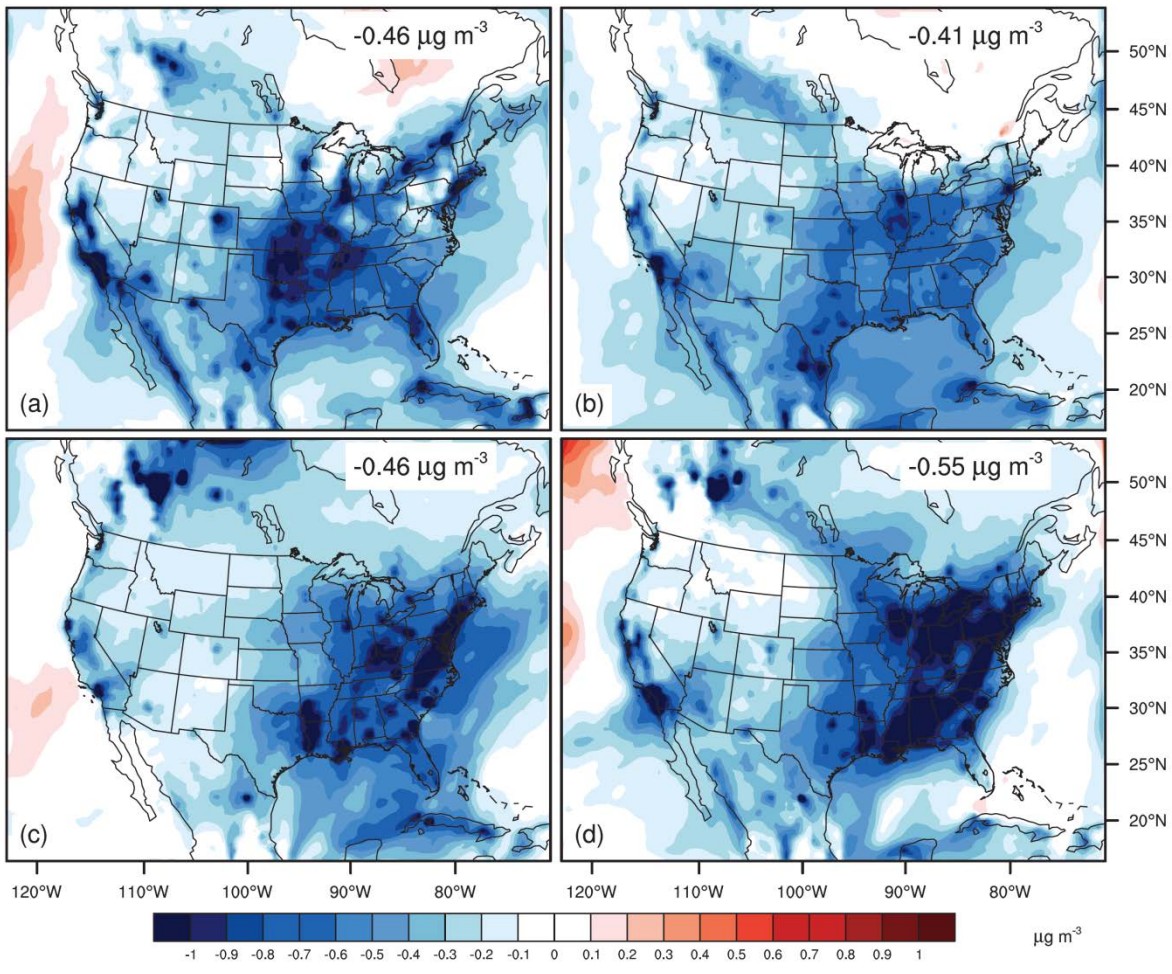

Fig. 4. Seasonal distributions of total co-benefits for PM$_{2.5}$ ($\mu$g m$^{-3}$) for (a) winter, (b) spring, (c) summer and (d) fall.



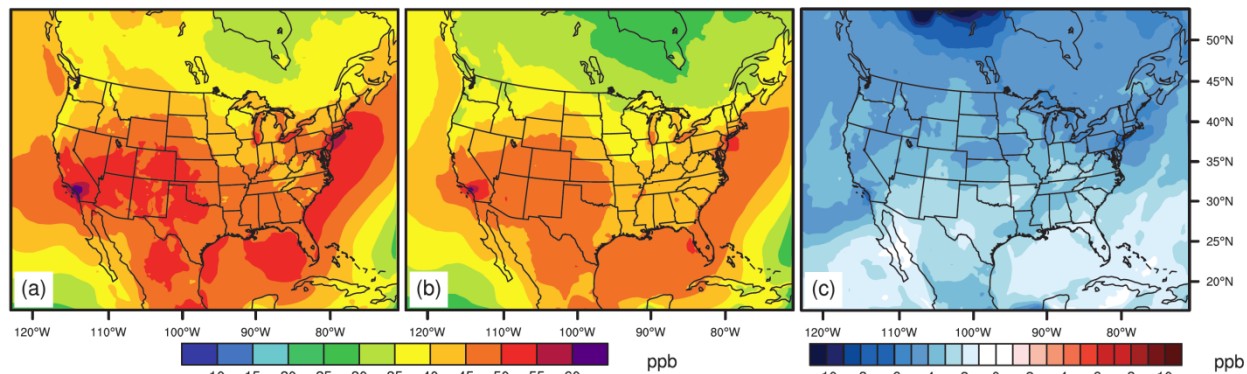

Fig. 5. The three-year ozone-season average (May to October) of MDA8 $O_3$ (ppb) from (a) S_ REF, (b) S_ RCP45, and (c) the total co-benefits (shown as the difference between S_RCP45 and S_REF). Blue colors in panel (c) indicate an air quality improvement.





Fig. 6. Benefits of reduced co-emitted air pollutants (a, b) versus slowing climate change (c, d) for PM$_{2.5}$ (a, c) and ozone season MDA8 surface O$_3$ (b, d). Blue colors indicate an air quality improvement. The numbers on the plots are the three-year average of air quality changes over the U.S.





Fig. 7. Benefits of domestic (a, b) versus foreign (c, d) GHG reductions for PM$_{2.5}$ (a, c) and ozone season MDA8 surface O$_3$ (b, d). Blue colors indicate an air quality improvement. The numbers on the plots are the three-year average of air quality changes over the U.S.




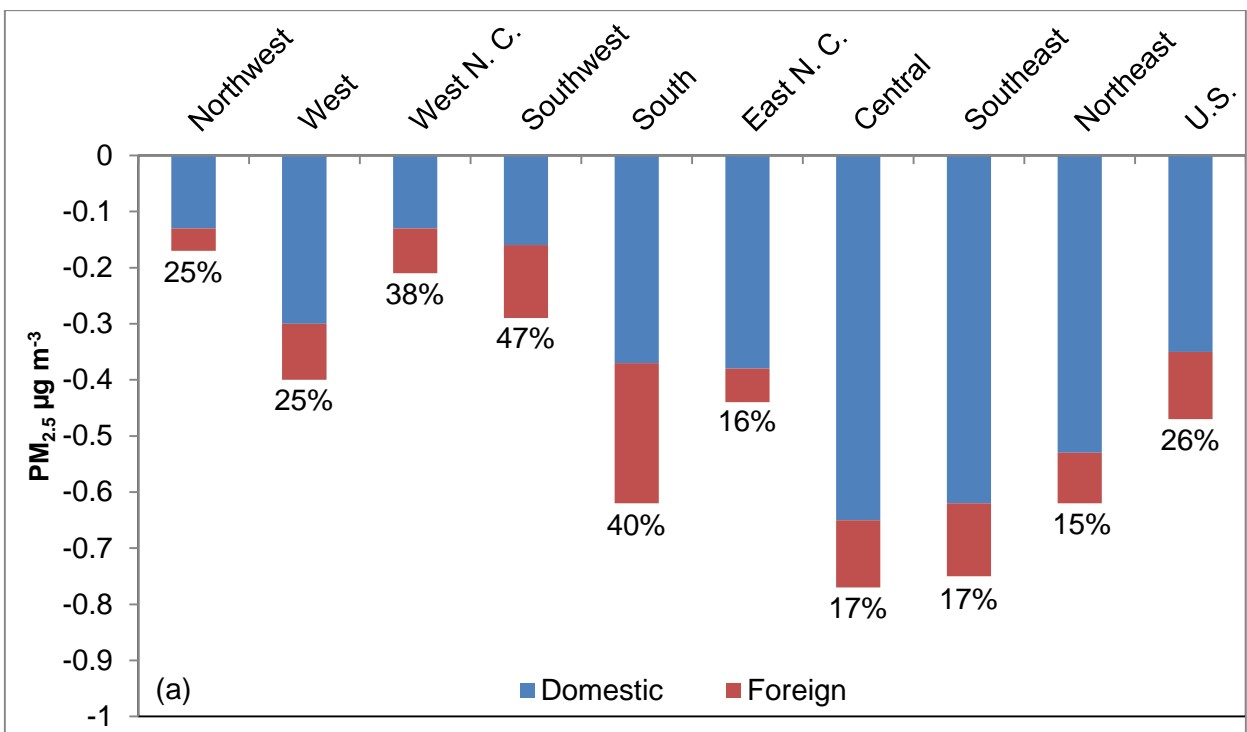

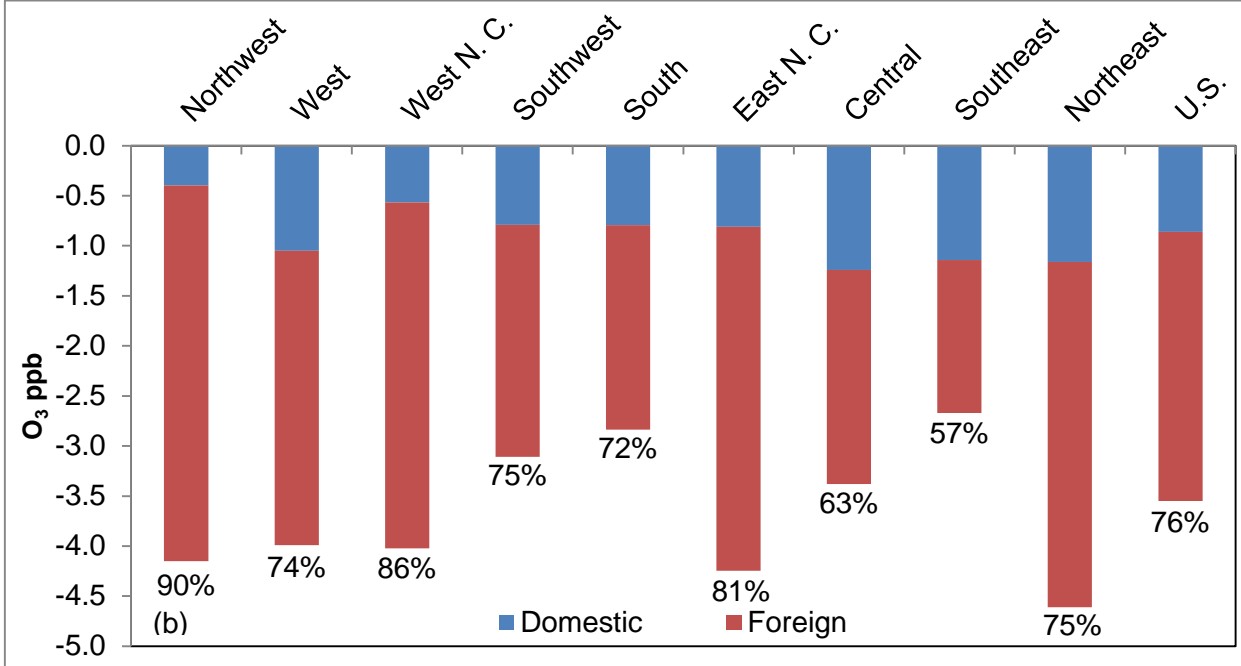

Fig. 8. Mean values of domestic (blue) and foreign co-benefits (red) for U.S. average (a) annual $PM_{2.5}$, and (b) ozone season MDA8 $O_3$. The numbers below each bar are the percentage (%) of the foreign co-benefit.