# Peer review of "Co-benefits of global and regional greenhouse gas mitigation on U.S. air quality in 2050"

_Atmospheric Chemistry and Physics, 2015_

## Referee Comment (RC1) · Y. Gao (Referee) · 19 Feb 2016

This study used dynamical downscaling to investigate the climate impact on air quality. Based on their previous work West et al. (2013), they applied relatively high resolution (36 km) to further explore some regional features of climate impact on O3 and PM2.5. The scenarios designed are interesting and I recommend its publication after addressing the concerns below:

Major comments:

1. The added values in regional models: In the world of dynamical downscaling, researchers try to improve model predictions using regional climate models despite intensive computational resources. It would be very useful if the authors show some comparisons between regional downscaled results and the driven GCM either in meteorology or air quality or both. For meteorology, the authors show changes in Figure S1 to S3, but we are not sure which one performs better. Similar question applies to air quality.

2. Figure S2 and Figure S3: the change in precipitation The values seem to be huge. The Figure S1 in the supporting information in Gao et al, 2014 (Gao et al., 2014, Robust spring drying in the southwestern U.S. and seasonal migration of wet/dry patterns in a warmer climate, Geophys. Res. Lett., 41, 1745–1751 ) shows the average annual precipitation of US is smaller than 5 mm/day (see the top left panel from that figure ). Mean summer precipitation change was shown in Figure 20 in the following paper (Eric D. Maloney, et al., 2014: North American Climate in CMIP5 Experiments: Part III: Assessment of Twenty-First-Century Projections. J. Climate, 27, 2230–2270), and mean winter precipitation was shown in Figure 1 in the following paper (Neelin et al., 2013: California Winter Precipitation Change under Global Warming in the Coupled Model Intercomparison Project Phase 5 Ensemble. J. Climate, 26, 6238–6256). Both of these two papers show increase of precipitation at about 1-2 mm/day from CMIP5 multi-ensemble mean results. The statement on Page 5, Line 25, "US average increase of precipitation 8.16 and 7.63 mm/day" is a rather large value. Please double check the WRF simulations. In addition, please look at historical precipitation, and see if it is comparable to PRISM (top left panel of the figure below; PRISM is an observational dataset). This comment also applies to Fig. 1 in the main paper showing large differences in precipitation between RCP 4.5 and RCP 8.5. Thus, to have a reasonable historical 3-year precipitation spatial distribution is essential before these comparisons make sense. Page 5, Line 29-30: "However, the only region where the regional climate is warmer and drier in RCP 4.5 is in the Northwest US". This statement does not make too much sense. Because RCP 4.5 mostly show smaller warming than RCP 8.5, which is expected, thus, the statement regarding "drier" should not combine with the warming statement. In particular, I am afraid this feature from a single model WRF may not be robust.

3. Section 3.5: Co-benefit from domestic and foreign GHG mitigation For domestic effect: S_Dom vs. S_REF (in Table 1). This is fine since the only differences between these two scenarios are emissions in US For foreign GHG mitigation: the authors compared S_RCP45 vs. S_Dom. I don't think these two scenarios can be interpreted as foreign GHG mitigation effect. Although the authors pointed out the limitations on Page 13 (Lines 11-15), I think the effect from US could be quite large. Since the authors did not conduct an experiment by reducing GHG over other countries only, the discussions related to foreign GHG mitigation should be revised (Figs. 7,8)

Minor comments:

1. There are quite a few places showing website link and accessed by a certain date. I suggest to move those links as footnotes and remove the words of "accessed date ***".

2. Page 8, Line 24-25 The authors claimed that US EPA (2007) recommended use median instead of mean for evaluation. In fact, I did not find this statement. US EPA (2007) does recommend NMB, MNB, and a few other metrics MFB, MFE. They also showed benchmarks (Page 252-261), although there is a range of the benchmarks. The biases (in %) shown in Table 3 is relatively small. However, I am not quite sure the evaluation of median is a good way. Did the authors look at the mean in other way if the authors do not prefer to show metrics with mean value, i.e., spatial patterns?

3. Page 8, Line 30 "emissions are derived from global datasets rather than specific emissions for US" The authors mentioned in section 2.2 that they used SMOKE to process emissions. I don't quite understand why the emissions are not specific for US.

4. Page 9, Line 15 "NOx titration" should be "NO titration" Figure S14: S_RCP45 is largely due to NO titration. However, in Figure S13, RCP 8.5 scenario or REF scenario, increase in methane concentration should play a big role in ozone increases. In fact, line 9-18 of Page 9 refers to the figures in the supporting information, which may not be an ideal way to present.
5. Page 10, line 25-26 and Table 1 for S_Emis and S_REF These two scenarios (S_Emis and S_REF) were used to evaluate the impact from emissions. Since in standard RCP 8.5, methane concentrations show dramatic increase which is closely related to ozone concentrations. My question is whether CH4 should keep the same as S_REF in S_Emis so as to be considered as RCP 8.5.

6. Fig. 6 (Page 29) should have titles for readers easy to recognize. The note of S_Emis-S_REF (described on Page 10, Line 26) is also useful to be added in the titles.

7. Page 13, Line 8 "only one model is used at each step during downscaling". This sentence needs to be rephrased. For example, simply removing "at each step" may work better.

---

## Referee Comment (RC2) · Anonymous Referee #2 · 26 Apr 2016

General Comments

The authors expand a previous study using dynamical downscaling and global emissions inventories in order to evaluate the impacts of global climate policy and climate change on regional air quality in the US, and compare these impacts to those modeled (in a consistent manner) at a coarser resolution. This study provides many valuable contributions including comparisons of: (1) regional vs global scale air quality co-benefits of consistent GHG emissions scenarios both with and without a climate policy (2) domestic vs foreign contributions of GHG emissions/policy to US air quality co-benefits (3) the effects of GHG policy on co-emitted pollutants vs changing climate on co-benefits. The study is ready for publication after a few minor revisions, assuming the units reported for the average change in rainfall amounts were a mistake.

Specific Comments

Given the importance of foreign GHG policies on US ozone, it would be helpful to see more information about the source of those rather large decreases in domestic ozone. There were two places where the paper suggested that global (but non-US) methane was the largest contributor to changes in domestic ozone, but it would be extremely helpful if that could be quantified, or discussed in more detail especially in light of the issue areas of the western US seems to be having with the idea of meeting more stringent ozone targets given these large contributions from "background" or "uncontrollable" sources. And if the reduction in methane concentration (as reported in Table 1) is indeed the largest source of the reduction of domestic ozone, what does that look like with respect to emissions? This finding was a big take-away from the paper and so more information about it from the policy perspective would be valuable.

Can you also clarify why US methane is not included? Even if inventory suggests it is small.

There seems to be some inconsistency between the original spatial distribution of emissions from the global inventories and the spatial allocation that was used for regional modeling of both emissions and meteorology. You mention that a benefit to using the global emission inventories (versus projecting the NEI) is that they take into account changing land use. But wouldn't both WRF and SMOKE use land use data that is both constant between 2000 and 2050, and inconsistent with the global representation?

Also, it is not clear to me how the emissions downscaling methods you used would provide any additional spatial detail (greater than that provided at a 0.5 x 0.5 degree level)? It seems that for VOCs and PM, there is detail added by scaling un-speciated totals by the speciation profile of the predominant source in each grid cell? But if I'm understanding your methods correctly, for any species other than VOC or primary PM there actually isn't any improvement in spatial allocation? If that is true, this is a downside you should mention as it would essentially smear your emissions out to the

global scale resolution. But perhaps I'm missing something, in which case, your paper would benefit from more clarity in this regard.

I was glad to see more details on the relative changes to different PM species as a result of the GHG policies and climate impacts, however, it was not clear why OM is the largest change? That would seem related to the changing climate, less so changing emissions, but changing emissions dominate so that doesn't explain what is going on with OM.

Technical Corrections

Page 5, line 25: That unit can't be correct. ?? Figure 1b has the same issue. Page 6, line 18-19: This sentence is not clear. Why not use the spatial allocation data available through SMOKE, or is that what you mean here? Page 8, line 24-26: it seems you used median for both ozone and PM2.5? Can you justify? Page 9, line 6: These are switched, are they not? Over-prediction is higher for MDA8? Page 13, line 5-7: Seems there is an error in this sentence.

---

## Author Comment (AC1) · 13 Jun 2016

Response to review #1 on acp-2015-1054

Co-benefits of global and regional greenhouse gas mitigation on U.S. air quality in 2050

Yuqiang Zhang, Jared H. Bowden, Zachariah Adelman, Vaishali Naik, Larry W. Horowitz, Steven J. Smith, and J. Jason West

We thank Referee #1 for providing thoughtful comments. We have responded to each comment below and have noted the page and line number for each revision to the manuscript. (blue colors are for referee's comments).

This study used dynamical downscaling to investigate the climate impact on air quality. Based on their previous work West et al. (2013), they applied relatively high resolution (36 km) to further explore some regional features of climate impact on O3 and PM2.5. The scenarios designed are interesting and I recommend its publication after addressing the concerns below:

The added values in regional models: In the world of dynamical downscaling, researchers try to improve model predictions using regional climate models despite intensive computational resources. It would be very useful if the authors show some comparisons between regional downscaled results and the driven GCM either in meteorology or air quality or both. For meteorology, the authors show changes in Figure S1 to S3, but we are not sure which one performs better. Similar question applies to air quality.

**Response**: We agree that the comparisons between the global models and downscaled regional models are very important to justify the downscaling work. We have added new analysis to the paper to provide more complete comparisons of both the global and regional models with observations. In Fig. S1, we add comparisons of 2-m temperature (T2) and precipitation for both the GCM and WRF results with observational data. We compare the T2 modeled by GFDL AM3 and WRF with 21 years of observations (1979 to 2000) from the North America Regional Reanalysis data (Mesinger et al., 2006). We also compare the precipitation from GFDL AM3 and WRF with 41 years of observations (1948 to 1998) from the Unified US Precipitation data products from NOAA Climate Prediction Center (Higgins et al. 2000). We see that the WRF downscaling helps to resolve important features that influence the average regional climate that are not resolved by GFDL AM3.

We rephrase the sentences in Pg 5 line 18-24 to show these changes:

"We compare the downscaled WRF and the global GFDL AM3 simulations (for three-year averages instead of four to be consistent with CMAQ outputs below), for 2-m temperature (T2) with 21 years (1979 to 2000) of observation data from the 32-km North America Regional Reanalysis (NARR; Mesinger et al. 2006), and for precipitation with 41 years (1948 to 1998) of observation data from the  $0.25^{\circ} \times 0.25^{\circ}$  Unified US precipitation data product from NOAA Climate Prediction Center (Higgins et al. 2000). The large-scale spatial patterns for both T2 and precipitation between WRF and GFDL AM3 are similar (Fig. S1). However, the downscaled simulations help resolve important features that influence the average regional climate, such as those related to topography."

For air quality, we added new comparisons between MOZART-4 and CMAQ for the simulated 2000  $PM_{2.5}$  and  $O_3$  (see the new Fig S11 in the supporting info). We see that the CMAQ is better in capturing the urban scale air quality than MOZART-4. We add the following text in Pg 9 line 7:

"By comparing the simulated annual  $PM_{2.5}$  and  $O_3$  in 2000 (both are three-year averages) between MZ-4 and CMAQ, we see that CMAQ captures urban scale air quality better than MZ-4 (Fig. S11)."

We also show in Fig 2 that the future PM2.5 changes for RCP4.5 in 2050 relative to 2000 (also see Fig S16 for comparison between REF in 2050 relative to 2000), are within the ACCMIP ensemble model means for both the MOZART-4 results and CMAQ results. For future O3 changes for RCP4.5 in 2050 relative to 2000, the results between MOZART-4 and CMAQ are similar. Spatially, we compare the total co-benefits for both PM2.5 and O3 between MOZART-4 and CMAQ in Fig S20-S25. As for the 2000 comparison, we see that CMAQ better simulates air quality changes in urban environments at finer scale. So we add one sentence in Pg 9 Line 27 "CMAQ better simulates air quality changes in urban environments at a finer scale compared with MZ-4."

Figure S2 and Figure S3: the change in precipitation The values seem to be huge. The Figure S1 in the supporting information in Gao et al, 2014 (Gao et al., 2014, Robust spring drying in the southwestern U.S. and seasonal migration of wet/dry patterns in a warmer climate, Geophys. Res. Lett., 41, 1745–1751) shows the average annual precipitation of US is smaller than 5 mm/day (see the top left panel from that figure ). Mean summer precipitation change was shown in Figure 20 in the following paper (Eric D. Maloney, et al., 2014: North American Climate in CMIP5 Experiments: Part III: Assessment of Twenty-First-Century Projections. J. Climate, 27, 2230–2270), and mean winter precipitation was shown in Figure 1 in the following paper (Neelin et al., 2013: California Winter Precipitation Change under Global Warming in the Coupled Model Intercomparison Project Phase 5 Ensemble. J. Climate, 26, 6238–6256). Both of these two papers show increase of precipitation at about 1-2 mm/day from CMIP5 multi-ensemble mean results. The statement on Page 5, Line 25, "US average increase of precipitation 8.16 and 7.63 mm/day" is a rather large value. Please double check the WRF simulations. In addition, please look at historical precipitation, and see if it is comparable to PRISM (top left panel of the figure below; PRISM is an observational dataset). This comment also applies to Fig. 1 in the main paper showing large differences in precipitation between RCP 4.5 and RCP 8.5. Thus, to have a reasonable historical 3-year precipitation spatial distribution is essential before these comparison s make sense. Page 5, Line 29-30: "However, the only region where the regional climate is warmer and drier in RCP 4.5 is in the Northwest US". This statement does not make too much sense. Because RCP 4.5 mostly show smaller warming than RCP 8.5, which is expected, thus, the statement regarding "drier" should not combine with the warming statement. In particular, I am afraid this feature from a single model WRF may not be robust.

**Response**: We thank the reviewer for calling our attention to this error and for providing detailed references. We became aware of this error soon after this paper was submitted, and found that it was due to our miscalculation of rainfall changes. We corrected this error in the new manuscript by replacing the plots in Fig 1 (b), Fig. S2(b) and Fig. S3(b). Please see Pg 5 Lines 25-26: "Additionally, precipitation is projected to decrease over most of the U.S. in both scenarios, with U.S. average decreases of 0.20 and 0.15 mm day-1 in RCP8.5 and RCP4.5." Also, as mentioned

in our first response, we compared WRF and GFDL AM3 annual average precipitation against CPC Unified Precipitation. We used the CPC precipitation instead of PRISM because of the horizontal resolution of CPC is of closer to WRF.

We deleted this text as suggested by the reviewer (Pg 5, Line 29-30): "However, the only region where the regional climate is warmer and drier in RCP 4.5 is in the Northwest US".

Section 3.5: Co-benefit from domestic and foreign GHG mitigation. For domestic effect: S\_Dom vs. S\_REF (in Table 1). This is fine since the only differences between these two scenarios are emissions in US For foreign GHG mitigation: the authors compared S\_RCP45 vs. S\_Dom. I don't think these two scenarios can be interpreted as foreign GHG mitigation effect. Although the authors pointed out the limitations on Page 13 (Lines 11-15), I think the effect from US could be quite large. Since the authors did not conduct an experiment by reducing GHG over other countries only, the discussions related to foreign GHG mitigation should be revised (Figs. 7,8) Response: We have constructed the scenarios to separate the total co-benefits (S\_RCP45-S REF) into components due to domestic emissions reductions (S Dom—S REF) and foreign emission reductions (S\_RCP45—S\_Dom). In comparing S\_Dom and S\_REF, the only difference is domestic air pollutant emissions. This provides a clean comparison to give an understanding of the importance of domestic emission changes, although it neglects the effect of US emissions on air quality through changes in global climate (via GHGs), regional climate (via short-lived climate pollutants), and methane. We consider that global climate and methane are mainly due to foreign emissions (as US emissions are much smaller than the rest of the world) and we neglect possible changes in regional climate from reductions in US short-lived climate pollutants.

We mainly focus on the effect of domestic reductions and compare that with the total. But we also find the effects of foreign reductions by simple subtraction  $(S\_RCP45\_S\_REF) - (S\_Dom\_S\_REF) = S\_RCP45\_S\_Dom$ . In doing so, we attribute the effects of all climate changes and methane changes to foreign reductions (as most GHGs and methane are from foreign sources). Doing more work to better understand changes in regional climate due to emission mitigation, and to attribute those changes to US emissions, would require extra simulations with WRF, SMOKE, and CMAQ; as these are computationally intensive, they are beyond what we can perform for this study.

We have improved the text in the Methods sections to better clarify the logic of our simulation design, and to better communicate its limitations (Pg 8 line 3):

"By comparing S\_Dom (applying GHG mitigation from RCP4.5 scenario in the U.S. only) with S\_REF, and S\_RCP45 with S\_Dom, we quantify the co-benefits from domestic and foreign GHG mitigation. The co-benefits from foreign reductions are found by simple subtraction  $(S_RCP45 - S_REF) - (S_Dom - S_REF) = S_RCP45 - S_Dom$ . In estimating the co-benefits of domestic reductions, we account for the influences of methane and of global climate change as foreign influences (as most methane and GHG emissions are outside of the U.S.), and assume that U.S. air pollutant emissions have small effects on global or regional climate, such as through aerosol forcing."

There are quite a few places showing website link and accessed by a certain date. I suggest to move those links as footnotes and remove the words of "accessed date \*\*\*".

**Response**: According to the ACP website:

http://www.atmospheric-chemistry-and-physics.net/for\_authors/manuscript\_preparation.html Footnotes: These should be avoided, as they tend to disrupt the flow of the text. If absolutely necessary, they should be numbered consecutively. Footnotes to tables should be marked by lowercase letters.)

We have chosen to keep these as they are. If the paper is accepted for publication, we will consult with the editor on whether our format is proper.

Page 8, Line 24-25 The authors claimed that US EPA (2007) recommended use median instead of mean for evaluation. In fact, I did not find this statement. US EPA (2007) does recommend NMB, MNB, and a few other metrics MFB, MFE. They also showed benchmarks (Page 252-261), although there is a range of the benchmarks. The biases (in %) shown in Table 3 is relatively small. However, I am not quite sure the evaluation of median is a good way. Did the authors look at the mean in other way if the authors do not prefer to show metrics with mean value, i.e., spatial patterns?

**Response**: The right reference should be from Appel et al. (2008) instead of the EPA (2007) report. We corrected in the main paper. Appel et al. (2008) suggested to use "median" over "mean" when the species evaluated are not normally distributed, which is common for PM species. They also suggested that if the data were normally distributed, the mean and median would be the same (section 4 paragraph 5 in Appel et al., 2008).

Page 8, Line 30 "emissions are derived from global datasets rather than specific emissions for US" The authors mentioned in section 2.2 that they used SMOKE to process emissions. I don't quite understand why the emissions are not specific for US.

**Response**: Here we mean that we are not using the NEI 2001 emission dataset, and instead downscale from the global RCP emissions in 2000. We rephrase this sentence to make it more clearly in Pg 8 Line 29-30:

"Model performance is not expected to be perfect as meteorology does not correspond with actual year 2000 meteorology, and emissions are derived from global datasets rather than the specific NEI dataset for the U.S."

Page 9, Line 15 "NOx titration" should be "NO titration" Figure S14: S\_RCP45 is largely due to NO titration. However, in Figure S13, RCP 8.5 scenario or REF scenario, increase in methane concentration should play a big role in ozone increases. In fact, line 9-18 of Page 9 refers to the figures in the supporting information, which may not be an ideal way to present.

Response: We changed from "NOx titration" to "NO titration". See Pg 9, Line 15.

We agree that methane increases in RCP8.5 plays an important role in winter  $O_3$  increases in the U.S., as suggested by the sensitivity simulation of Gao et al., (2013). We added new sentences in Pg 9, line 15-16:

"O3 increases over the Northeast and West U.S. in winter in both S\_REF and S\_RCP45, caused by the weakened NO titration as a result of the large NO decrease in the two scenarios (Table 2), as also reported by other studies (Gao et al., 2013; Fiore et al., 2015), and as well as the large methane increases in RCP8.5 (Gao et al., 2013)." With respect to the fact that Figures S12-S15 are in the supporting information, we have chosen to emphasize the co-benefits (Section 3.3) in this paper and so have included figures addressing the co-benefits in the main paper (Figs. 3-5). The changes relative to the year 2000 (Section 3.2) are included in the paper mainly as backdrop for understanding the future simulations, and for comparison against other studies that simulated 2050 relative to 2000 for these same scenarios. We appreciate the reviewer's suggestion to put one or more of these figures in the main paper, but have chosen to keep them in the supporting information so that the main paper will not be excessively long.

Page 10, line 25-26 and Table 1 for S\_Emis and S\_REF. These two scenarios (S\_Emis and S\_REF) were used to evaluate the impact from emissions. Since in standard RCP 8.5, methane concentrations show dramatic increase which is closely related to ozone concentrations. My question is whether  $CH_4$  should keep the same as S\_REF in S\_Emis so as to be considered as RCP 8.5.

**Response**: By comparing between S\_REF and S\_Emis, we want to see the effect of global emission reductions from the RCP4.5 scenario, separate from changes in climate. These are the two mechanisms of co-benefits identified and quantified by West et al. (2013). We use methane concentrations from RCP4.5 in S\_Emis to simulate the effect of all emission reductions, including both methane and short-lived air pollutants. The reviewer asks about RCP8.5, but we use the REF scenario for emissions throughout the paper and not RCP8.5. This design of scenarios (Table 1) is consistent with the global simulations of West et al., (2013). We have added text to the discussion of Table 1 to clarify that methane decreases in S\_Emis (Page 8, Line 1-3): "The emission benefit from the first mechanism is calculated as the difference between S\_Emis and S\_REF, for which the change in methane concentration is included as an emission benefit, and the meteorology benefit is calculated as S\_RCP45 minus S\_Emis."

Fig. 6 (Page 29) should have titles for readers easy to recognize. The note of S\_Emis-S\_REF (described on Page 10, Line 26) is also useful to be added in the titles. **Response**: We added the notes of "S\_Emis-S\_REF" and "S\_RCP45-S\_Emis" into Fig 6. We also added the notes in Fig 7.

Page 13, Line 8 "only one model is used at each step during downscaling". This sentence needs to be rephrased. For example, simply removing "at each step" may work better. **Response**: We have revised text in Pg 13 line 8 as:

"Moreover, only one model is used during downscaling for regional climate (WRF) and air quality (CMAQ) modeling, and the mean of a model ensemble can be used to reduce model error."

---

## Author Comment (AC2) · 13 Jun 2016

Response to review #2 on acp-2015-1054

Co-benefits of global and regional greenhouse gas mitigation on U.S. air quality in 2050

Yuqiang Zhang, Jared H. Bowden, Zachariah Adelman, Vaishali Naik, Larry W. Horowitz, Steven J. Smith, and J. Jason West

We thank referee #2 for the positive and constructive suggestions and comments, which have helped us improve the manuscript. All comments have been carefully addressed here (blue colors are for referee's comments), and we have tracked all changes in the revised manuscript.

The authors expand a previous study using dynamical downscaling and global emissions inventories in order to evaluate the impacts of global climate policy and climate change on regional air quality in the US, and compare these impacts to those modeled (in a consistent manner) at a coarser resolution. This study provides many valuable contributions including comparisons of: (1) regional vs global scale air quality co-benefits of consistent GHG emissions scenarios both with and without a climate policy (2) domestic vs foreign contributions of GHG emissions/policy to US air quality co-benefits (3) the effects of GHG policy on co-emitted pollutants vs changing climate on co-benefits. The study is ready for publication after a few minor revisions, assuming the units reported for the average change in rainfall amounts were a mistake.

**Response:** The reviewer is correct that the rainfall changes were wrong by miscalculation, and we have corrected this error in the new manuscript, Fig 1 (b), Fig. S2(b) and Fig. S3(b). Please see Pg 5 Lines 25-26: "Additionally, precipitation is projected to decrease over most of the U.S. in both scenarios with U.S. average decrease of 0.20 and 0.15 mm day$^{-1}$ in RCP8.5 and RCP4.5."

Given the importance of foreign GHG policies on US ozone, it would be helpful to see more information about the source of those rather large decreases in domestic ozone. There were two places where the paper suggested that global (but non-US) methane was the largest contributor to changes in domestic ozone, but it would be extremely helpful if that could be quantified, or discussed in more detail especially in light of the issue areas of the western US seems to be having with the idea of meeting more stringent ozone targets given these large contributions from "background" or "uncontrollable" sources. And if the reduction in methane concentration (as reported in Table 1) is indeed the largest source of the reduction of domestic ozone, what does that look like with respect to emissions? This finding was a big take-away from the paper and so more information about it from the policy perspective would be valuable.

**Response:** We agree that it would be useful to quantify the influence of global methane changes under RCP4.5 on U.S. ozone with extra sensitivity simulations. However doing so will require more simulations of both global and regional models. Considering the intense computational requirement, we are unable to perform these extra model simulations here. We suspect that methane emissions are very important because there is a large difference in methane between REF and RCP4.5. For the global co-benefits study (West et al., 2013), we tried to estimate the methane contribution by multiplying the change in methane by the present-day sensitivity of ozone to methane, which yielded a very large ozone change that was greater than the total change. The problem is that 2050 precursor emissions are lower than at present, and so the sensitivity to methane would also be lower. Without extra simulations, we cannot quantify this effect.

Methane is included in both the global and regional model as a fixed global concentration, but at the Referee's request, we have reported the global anthropogenic emissions that these correspond to (Pg 11 Line 11): "This large influence of foreign reductions for $O_3$ highlights the importance of global methane reductions in RCP4.5 (anthropogenic emissions of 330 Tg yr$^{-1}$ in 2050 in RCP45, compared to 432 Tg yr$^{-1}$ in REF), and air pollutant emission reductions particularly in Asia and intercontinental transport.

Can you also clarify why US methane is not included? Even if inventory suggests it is small.
**Response:** The US methane emissions are only a small fraction of the global $CH_4$ emissions (32 Tg yr$^{-1}$ in REF in 2050, 7.4% of the anthropogenic total). We didn't model the effects of methane emissions directly, but instead use fixed concentrations of methane in 2050 that correspond with the emissions in the different scenarios. It would be possible to calculate the change in global methane concentration associated with US methane emissions alone (or foreign emissions alone), but that calculation would be uncertain. Doing so would also require additional simulations with the global model MOZART-4 in addition to CMAQ. Instead we chose to treat methane emissions as entirely from foreign sources and acknowledge the small error involved.
To address the reviewer's concern, we rephrase the sentence in Pg 8 line 7-9:
"In each scenario, we fix global methane at concentrations given by the RCPs (Table 1), and account for methane changes as a foreign influence, neglecting the fraction of global anthropogenic methane emissions that are from the U.S. (7.4% in 2050 REF scenario and 7.0% in 2050 RCP4.5)."

There seems to be some inconsistency between the original spatial distribution of emissions from the global inventories and the spatial allocation that was used for regional modeling of both emissions and meteorology. You mention that a benefit to using the global emission inventories (versus projecting the NEI) is that they take into account changing land use. But wouldn't both WRF and SMOKE use land use data that is both constant between 2000 and 2050, and inconsistent with the global representation?

Also, it is not clear to me how the emissions downscaling methods you used would provide any additional spatial detail (greater than that provided at a 0.5 x 0.5 degree level)? It seems that for VOCs and PM, there is detail added by scaling un-speciated totals by the speciation profile of the predominant source in each grid cell? But if I'm understanding your methods correctly, for any species other than VOC or primary PM there actually isn't any improvement in spatial allocation? If that is true, this is a downside you should mention as it would essentially smear your emissions out to the global scale resolution. But perhaps I'm missing something, in which case, your paper would benefit from more clarity in this regard.
**Response:** We address these related comments together. The reviewer is right that we use constant (year 2000) land use and land cover in both the WRF and CMAQ simulations. This would lead to inconsistencies in the land use and land cover distributions and the spatial distribution of emissions. For CMAQ this would be important, for example, for biogenic VOC emissions. We are not aware of any downscaling study that simulates spatial distributions land use changes and anthropogenic emissions in a way that is completely consistent, and we are planning to do work that increases this consistency in the future.

For anthropogenic emissions, however, we account for future land use changes and their effects on the spatial distribution of emissions in the two scenarios. In the paper, we contrast our emission downscaling method with the traditional NEI scaling method, which multiplies the spatial emissions from the current NEI by a national mass ratio between the future emissions and the current NEI. By doing this, the traditional method assumes that the air pollutant spatial distributions in the future stay the same as the current NEI. In contrast, we use the projected RCP emissions datasets that include projected changes in the spatial distribution of emissions at $0.5 \times 0.5$ degree resolution, and regrid those to the CMAQ grid (36 km). Because we simply regrid the RCP emissions, our methods do not provide additional spatial detail beyond what is provided by the RCPs at 0.5 degree resolution. Our results show that the spatial distribution of emissions do change in the future (rightmost plots in Figs. S4-S10).

We have added text in Pg 6 line 10 to the paper to acknowledge the inconsistency in land use assumptions:
"We use constant (year 2000) land use and land cover for all simulations in WRF and CMAQ, whereas the spatial distributions of anthropogenic emissions change in the RCP scenarios."

To clarify, we add text in section 2.2 Pg 6 line 4:
"By doing this, the traditional method assumes that future spatial distributions of emissions stay the same as the current NEI."

We rephrase text in Pg 6 line 9-10:
"By regridding the REF and RCP4.5 data, we account better for changes in the spatial distribution of future emissions projected in the RCPs (Figs. S4-S10), but do not provide additional spatial detail beyond what is provided by the RCPs at 0.5 degree resolution,"

I was glad to see more details on the relative changes to different PM species as a result of the GHG policies and climate impacts, however, it was not clear why OM is the largest change? That would seem related to the changing climate, less so changing emissions, but changing emissions dominate so that doesn't explain what is going on with OM.
**Response:** The OM decreases are dominated by the decrease of primary organic carbon (POC, 0.074 µg m$^{-3}$ decreases). By adding new figures showing the OM component changes from emission reductions and climate (Fig S19 in the supporting info), we see that the POC decreases are mainly caused by emission reductions, and the SOA decreases by changing climate. To clarify, we add the following text in Pg 10 Line 30 "In Fig. S18, the OM decrease is caused mainly by primary organic carbon (POC, 0.074 µg m$^{-3}$ decreases), followed by biogenic SOA (ORGB, 0.057 µg m$^{-3}$) and non-carbon organic matter (NCOM, 0.048 µg m$^{-3}$). The POC and NCOM decreases are caused mainly by emission reductions, while the SOA decrease is caused mainly by changing climate (Fig. S19)."

Page 5, line 25: That unit can't be correct. ??
**Response:** We corrected magnitude change for precipitation in the manuscript, as well as Fig. 1(b), Fig. S2(b) and Fig. S3(b).

Page 6, line 18-19: This sentence is not clear. Why not use the spatial allocation data available through SMOKE, or is that what you mean here?

**Response:** This sentence refers to our method of back-calculating total PM emissions from BC and OC, and does not refer to the spatial allocation of emissions. When we downscale the RCP emissions for CMAQ, there are more than one sub-categories inside one sector, e.g., the sector "Industries" includes emissions from "1A2_2A_B_C_D_E" as listed in Table S1. SMOKE provides more detailed sub-category information for the emission sectors. So when we match the emission sectors between the global RCPs and SMOKE, there usually are more than one category, and the speciation cross-reference files are slightly different between each sub-category. When that happens, we use the speciation cross-reference file from the sub-category with largest mass fraction in this sector, following the methods of Reff et al. (2009) and Xing et al. (2013).

To make it more clearly, we revise the sentence in Pg 6 line 18-19:
"In back-calculating total PM emissions from BC and OC, there are usually more than one sub-category within one sector, e.g., the sector "Industries" includes emissions from the sub-category of "1A2_2A_B_C_D_E" (Table S1). When that happens, we use the speciation cross-reference file from the sub-category with largest mass fraction in this sector, following the methods of Reff et al. (2009) and Xing et al. (2013)."

Page 8, line 24-26: it seems you used median for both ozone and PM2.5? Can you justify?
**Response:** From a recent CMAQ evaluation paper (Appel et al., 2008), they suggested using "median" over "mean" when the species evaluated are not normally distributed, which is commonly the case for PM species. They also suggested if the data were normally distributed, the mean and median would be the same. We provided the right reference for this sentence in the new manuscript.

Page 9, line 6: These are switched, are they not? Over-prediction is higher for MDA8?
**Response:** Yes, and it should be "The overprediction is slightly lower for 1hr-$O_3$ than for MDA8-$O_3$". We fixed in the new manuscript (Pg 7, line 6).

Page 13, line 5-7: Seems there is an error in this sentence.
**Response:** We used spectral nudging in our WRF downscaling studies, and meant to compare here with "analysis nudging". We updated in the new manuscript Pg 13 Line 5-7:
"Spectral nudging is adopted in this study to restrain WRF from drifting from the GCM, which has been shown to be better for some meteorological variables, but analysis nudging better for others (Bowden et al., 2012, 2013; Liu et al., 2012; Otte et al., 2012)."